# The Visibility of Citizen Participation and the Invisibility of Groundwater in Mexico

Ojilve Ramón Medrano-Pérez [1], Luzma Fabiola Nava [1,2,*] and Antonio Cáñez-Cota [3]

1   CONACYT-Centro del Cambio Global y la Sustentabilidad, A.C. (CCGS), Villahermosa 86080, Mexico; omedrano@conacyt.mx
2   International Institute for Applied Systems Analysis (IIASA), A-2361 Laxenburg, Austria
3   CONACYT-Centro de Investigaciones y Estudios Superiores en Antropología Social (CIESAS), Guadalajara 44190, Mexico; acanez@conacyt.mx
*   Correspondence: lnava@conacyt.mx or nava@iiasa.ac.at

**Abstract:** The aim of this study is to assess the social responses to protect and prevent conflict over groundwater resources. By means of a qualitative method and a study of the Valles Centrales and Valle de Mexicali aquifers in Mexico, we found that centralized water management, citizens' socio-environmental awareness, an asymmetry of power between stakeholders, an imposition of government policies, and economic interests are all contributory factors to emerging conflicts over groundwater. However, citizen participation has developed to provide organized individuals with an opportunity to influence public decisions through the recognition of their rights with respect to water inequalities. However, a limitation of the study is the illustration of conflictual events through the interpretation of qualitative data and of the opinions of the actors studied. However, the construction of hydrosocial territory in these aquifers is concretized in the potentiality and significance of citizen participation in promoting sustainable and socially responsible public groundwater policy at the regional level.

**Keywords:** aquifer; adaptation; innovation; management; political will; sustainability

## 1. Introduction

Groundwater is an invisible resource: however, it has rendered visible the social, economic, political, and environmental conflicts in particular areas [1–3]. The field of political ecology involves the study of the power relations articulated by different actors with respect to natural resources or common goods. It also contributes to an interdisciplinary understanding of the ecological and distributive conflicts and disputes generated by the dominant economic system [4,5]. Political ecology allows us to understand how nature and culture resist the standardization of socioeconomic, environmental, and political-institutional values and processes that tend to be dominated by market values in particular territories [6].

The concept of territory reveals a set of social relations surrounding water. An emphasis on the role of water in the production and reproduction of power sees territory as hydrosocial [7]. The analysis of hydrosocial territory from the perspective of political ecology evokes the territorial processes through which we can understand the socialization of the hydrological cycle within the diverse geometries of power that operate within it, that condition the forms of appropriation, distribution, and use of water, and that affect the behavior of specific social groups through their access to the resource [8,9].

The concept of hydrosocial territory allows for the reconsideration of water scenarios and the understanding of challenges imposed by its appropriation, distribution, and use [8]. For Swyngedouw [10], the concept of hydrosocial territory reflects how the flow of water, capital, and power are materially connected at different levels of political, economic, social, and ecological factors. For Boelens et al. [11], hydrosocial territory suggests that a diagnosis

of environmental, technological, economic, social, political, and institutional interactions allows us to understand the social disputes, conflicts, and responses surrounding water. According to Boelens et al. [11] and Damonte and Boelens [12], human practice, the flow of water, hydraulic technology, the biophysical environment, and socioeconomic, institutional, and cultural structures that interact in the control of water, are configured in this territory. Pelayo and Gasca [13] argue that hydrosocial territory is that which allows us to outline how regions with water articulate the productive and reproductive dynamics of communities, including the political course of water policy, the effect of public and private interventions, and conflicts among parties for the appropriation, use, and control of water.

It is against this background that water resources convey the relationships between actors in a hydrosocial territory [14]. Continuous and dynamic social processes, by means of which water and this ensemble of heterogeneous actors integrate and constitute themselves in time and space [15], favor the origin of complex and varying socio-spatial configurations [10,16–19], especially in areas where water is scarce [20]. Social relations within this territory end up transforming it. That is, they humanize it [11,14,19] and instrumentalize it for a multidimensional analysis of territorial and social transformations [12].

The aim of this study is thus to assess the social responses for the defense of, and prevention of conflict over, groundwater resources in the following two case studies: the Valles Centrales aquifer in Oaxaca and the Valle de Mexicali aquifer in Baja California. It asks how citizen participation in each of these regions has favored reaching a consensus on managing groundwater. We argue that citizen participation in both regions is characterized by a mosaic of relations—environmental, social, cultural, political, and economic—constructed by multiple social groups that relate to one another around groundwater to satisfy their needs and interests, and that they organize in defense of the territory and to protect their resources. The construction of hydrosocial territory is concretized in the potentiality and significance of citizen participation to promote sustainable and socially responsible public groundwater policy at the regional level.

## 2. Groundwater in Mexico

The three-fold problem of the spatial-temporal availability of water, the distribution of the population, and of economic development presents a challenge to water management and socio-environmental benefits [21–28]. A particular difficulty is an increasing deterioration of the available groundwater. The origin of this deterioration traces back to the year 1945, with the modification of Article 27 of the Mexican Constitution, which specified federal authority in its exploitation and management [29]. Since that time, the governmental logic in territorial legislation and management has been characterized by centralist authoritarianism bereft of measures to conserve or protect the resource [27,30,31]. Indeed, the number of overexploited aquifers rose from 32 in 1972 to 157 in 2020. The increasing degradation of groundwater, the growing competition in use, and the challenges imposed by the human right to water and sanitation and to a healthy environment are factors behind such conflicts, to which affected communities have responded to with organization and participation.

In Mexico, the major consumer of water is agriculture, which uses more than 70% of the resource [32]. Such use requires a concession, which establishes the user's right to a specific volume for a specific purpose during a specified period of five to 30 years, according to Article 24 (Chapter 2) of the National Water Law (Ley de Aguas Nacionales, LAN). The concession is recorded in the Public Register of Water Rights (Registro Público de Derechos de Agua, REPDA), which is administered by the National Water Commission (Comisión Nacional del Agua, CONAGUA). According to the REPDA, there were 516,396 water concessions as of 31 May 2019, with a total consumption volume of 87,400 Hm$^3$: of which 75% is used for agriculture, 15.3% for the public water supply, 4.8% for industry, and 4.9% for energy (not counting hydroenergy). Although, the major source of the concession is surface water (60%), Mexico is among the ten major consumers in the world of groundwater [33]. Moreover, the predicted effects of climate change will lead to a decline in precipitation in

the northern and central regions of the country, while in the southeast, the spatio-temporal availability of water will intensify and lead to flood impacts [34,35].

### 3. Materials and Methods

This study used qualitative methods to recognize the significance of citizen participation in the process of decision making and innovations in water policy. Social relations matter, and the words and actions of the actors studied are of great importance. A review of the literature relative to social participation and conflict surrounding groundwater management and in Mexico was the focus of a descriptive and contextual approach based on what stakeholders collectively believe to be the source of the conflictual situation, and what they wish for the future in terms of their relationship to water resources in the short, medium and long terms.

The purpose of this exercise was to describe the data without conceptualizing or interpreting them, in order to faithfully illustrate the situation in question. Detailed information (scientific, technical, and official documents, as well as online newspaper archives), based on the most recent information, accountability, and coherence, was collected and analyzed to develop a holistic vision of the object under study [36]. A case study method was also used to fully understand it in its real context, simultaneously using multiple sources of evidence [37]. These actions imply a process of investigation characterized by a deep and systematic examination of an event, understood as a unique social entity [38]. That is, the case study allows us to document experiences [39] and to obtain a global image of the interaction of the factors that influence the event [40]. The choice of the two case studies has a theoretical character, based on an identification of the key element or variables influencing the event, and on an understanding of the event, that considers all relevant variables [41,42].

Each of the cases analyzed were divided into subunits of analysis [43]. The case studies of the Valles Centrales and Valle de Mexicali aquifers (Figure 1) are two paradigmatic examples of the present-day challenges and opportunities of citizen participation in the problematic management of groundwater. Both represent recent contexts with diverse actors and characteristics that together exemplify the growing conflict. Valles Centrales was selected because of the existing research studies provided in the literature [27] that analyze the role of citizen organizations in the creation of the regulatory framework for access to and in the management of the aquifer (Figure 1a). Valle de Mexicali was selected because it is the object of the most recent bout of national media coverage of the problem, but with different actors and characteristics (Figure 1b). In both cases, however, citizen participation is a key and common factor in the case study analysis.

The methodological contribution of the study is twofold. First, it examines the patterns in the behavior of citizen participation regarding conflicts over groundwater. Second, it contributes to the debate about water policy with evidence obtained from the case studies, by objectively identifying the facts just as they presented themselves. With these ideas in mind, it is important to highlight that two main roots of methodological strategies in the literature of citizen participation have been identified. First, an evaluation of the effectiveness of citizen participation; that is, research focusing on explaining the extent to which stakeholders improve societal outcomes and achieve public value [44]. Second, a search of the intersection between citizen participation and policy innovation, focusing on agenda setting [45]. That is, increasing the legitimacy of decision-making processes towards collective learning, since some democracies have weak institutions by which to enforce accountability [46]. Our approach is based on this last factor, since our methodological strategy is based on describing citizen participation that aims to block imposed water policies and to promote innovation by opening avenues of decision making for a more general form of participation.

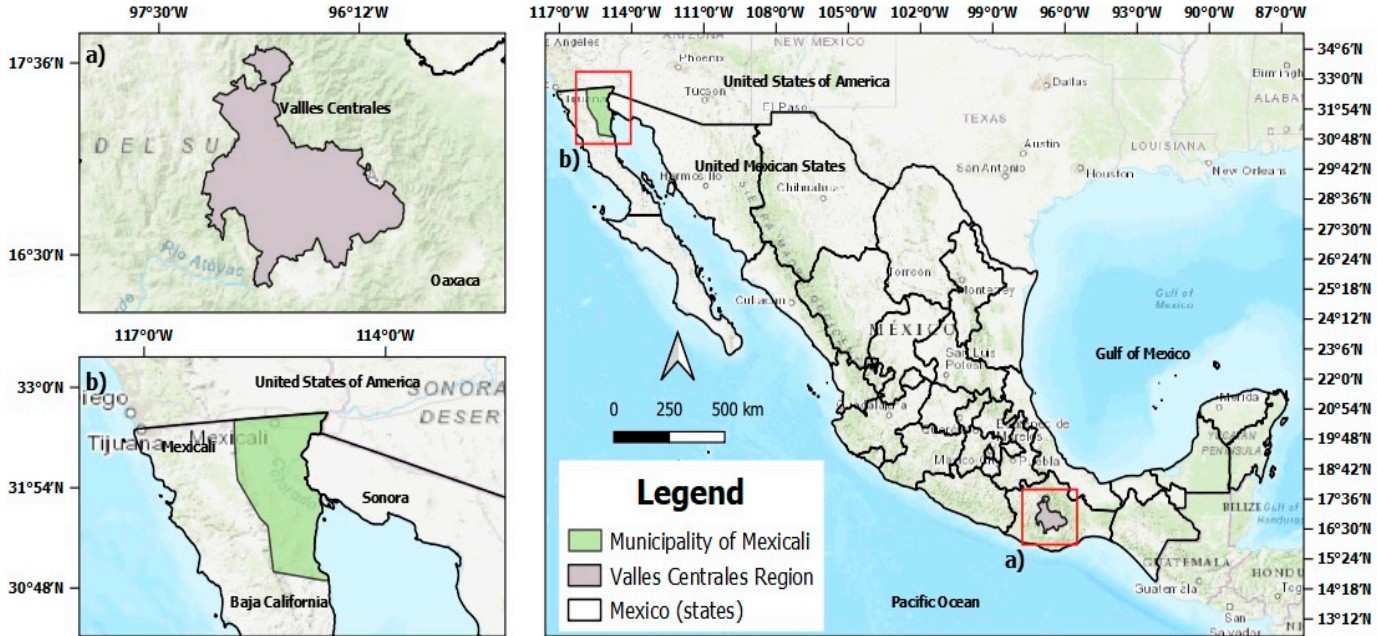

**Figure 1.** Location of the areas studied: (**a**) Valles Centrales region (Oaxaca); and (**b**) Municipality of Mexicali (Baja California). Source. Created with Qgis v3.14.16, with data from the Instituto Nacional de Estadística y Geografía (INEGI).

Nevertheless, the methodological limitations of the study are reflected in the illustration of events through sources of information. This means that even if qualitative data cannot be objectively measured or counted, it expresses the interpretive qualities of an event. Then, this event, associated with ideas, opinions, values, and behaviours of individuals, makes room for the researcher to explore and provide deeper insights into real-world problems. This said, a qualitative method has been used to understand people's beliefs, experiences, attitudes, behavior, and interactions with the aim of looking at meaning, perspectives and motivations [47].

Therefore, we propose a future research agenda that involves conducting in-depth interviews and focus groups for a more comprehensive qualitative research design and to understand the motives and actions of the actors involved. This strategy will help to identify subjectivities that can further develop a discussion of the results in a more complex and robust way and, can provide tools to learn more about causal processes in citizen participation in groundwater projects.

### 3.1. Valles Centrales, Oaxaca

The Valles Centrales region includes the capital of the state of Oaxaca and is known also as Los Valles or Valle de Oaxaca [27]. This valley, lying on the Valles Centrales aquifer, is in the central region of Oaxaca, one of the state's eight regions (Figure 2). Valles Centrales consists of 121 municipalities that are organized into seven districts (Ocotlán, Zimatlán, Zaachila, Etla, Ejutla, Tlacolula, and Centro) and 1280 localities. According to the State Commission for the Development of Oaxaca (Comisión Estatal para el Desarrollo de Oaxaca, COPLADE-OAX), the region has an area of 9480 km$^2$ and in 2010 the population numbered 1,107,557 residents, representing approximately 29% of the state's population. Approximately 25% of the population lives in small, dispersed, rural localities, 40.9% in cities, and 33.2% in communities that are in transition from rural to urban [48]. The climate varies from semi-warm subhumid in the plains to temperate subhumid in the high mountain elevations. The average annual summer rainfall ranges from 600 to 730 mm [49,50]. Approximately 50% of the population identifies as indigenous, mainly Zapotec, Mixtec,

and Mixe. The major economic activities are tourism, agriculture, and, to a lesser extent, industry; these activities constitute the major use of water in the region.

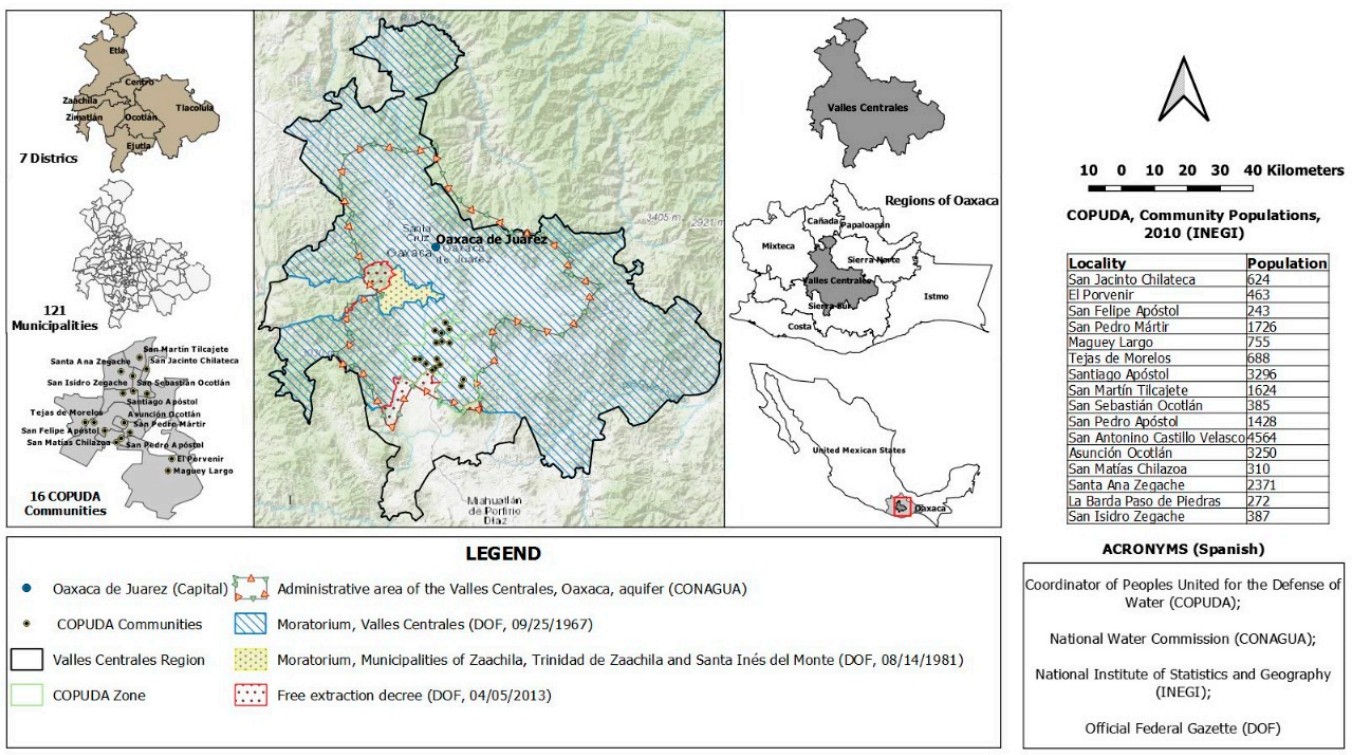

**Figure 2.** Location of the Valles Centrales region, Oaxaca, and of the Valles Centrales Aquifer. Source. Created with Qgis 3.14.16, with data from the Instituto Nacional de Estadística y Geografía (INEGI).

Water resources in the region mainly originate from subterranean sources. Deep wells provide the major source of water for human consumption [51]. The region consumes 121.8 Mm$^3$ of water annually [52]. Around 90% of the water for human use and consumption is taken from deep wells, 8% from water wheels, and 2% from springs and infiltration galleries [53]. Groundwater provides about 40% of the agricultural demand, in addition to supplying the industrial sector [54]. In the Valles Centrales the Río Atoyac, a tributary of the Río Verde is the major source of replenishment of the Valles Centrales aquifer [50].

The Valles Centrales aquifer is located in Hydrological Region 20, Costa Chica of Guerrero, in Subregion 20B, Costa Chica-Río Verde, Río Atoyac Basin. It shares a border with the watershed of the Río Atoyac-Oaxaca de Juárez Sub-basin. It has an area of 3769.40 km$^2$ and an extension of 5940 km$^2$, with an extraction zone of approximately 1130 km$^2$ [55]. The average availability of groundwater in the Valles Centrales aquifer is 12.61 Hm$^3$ year$^{-1}$, and the volume of extraction is 122.6 Hm$^3$ year$^{-1}$ [56] The total volume of the concession is 121.7 Hm$^3$ year$^{-1}$, and the volume of replenishment is 153.6 Hm$^3$ year$^{-1}$. The major uses are for agriculture and urban public services, and to a lesser extent, industry and tertiary services [55].

According to the REPDA database, the state of Oaxaca reported a total of 32,663 concessions as of 31 May 2019, equivalent to a total volume of 1462 Hm$^3$ year$^{-1}$, not counting the 16,800 Hm$^3$ year$^{-1}$ concession to hydro-energy. Of this amount, 77% was distributed to agriculture, 18% to urban public services, 2% to industry, and 3% to other uses. Surface water was the major source, providing 71% of the total concession, and the remaining 29% was from groundwater. In the Valles Centrales region there were 9972 concessions, with a total volume of 388 Hm$^3$ year$^{-1}$, which is 27% of the state's total; 60.4% went to agriculture, 38.1% to urban public services, and 1.5% to industry, not country hydro-energy.

The state of Oaxaca has historically had problems of water supply, with the epicenter of the problem in the capital city. According to González et al. [57], the major water problems

include lack of availability, deterioration of the distribution infrastructure, overexploitation of aquifers, and contamination from leaks in sewers and the discharge of industrial and municipal water, as well as misadministration, inequitable distribution, and irrational use. The Valles Centrales region has also been affected by climate change, which has caused water shortages and damage to agriculture [58]. However, the conflicts in Valles Centrales over access to water have been mainly political-institutional and socioeconomic, in addition to conflict over deficient infrastructure [27,59].

### 3.2. Valle de Mexicali, Baja California

The municipality of Mexicali is the capital of the state of Baja California, located on the Baja California peninsula. According to the Baja California State Development Commission [60], the municipality has an area of approximately 14,541 km$^2$, with an estimated population of 1,091,604, in 2020, comprising close to 30% of the state's population. Agriculture and geothermal energy, both intensive users of water, are two of the major economic activities in this region. Water in the region originates from subterranean sources and from the Colorado River (Figure 3).

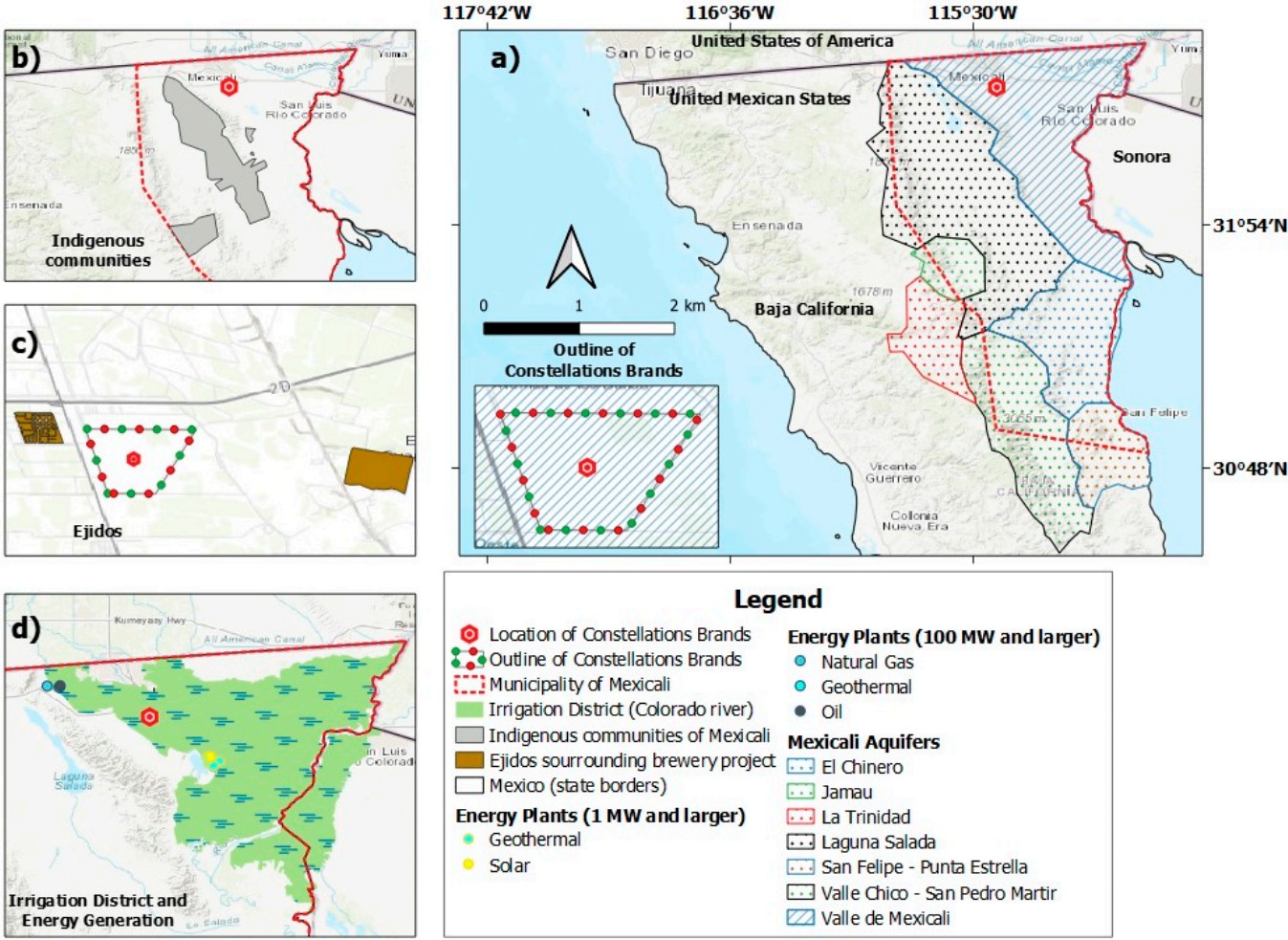

**Figure 3.** Location of Valle de Mexicali, Baja California. (**a**) Location of the AVM and Constellations Brands; (**b**) Municipality of Mexicali and location of indigenous communities; (**c**) Ejidos surrounding brewery project;(**d**) location of Irrigation District and Energy Generation sites. Source. Created with Qgis 3.14.16, with data from the Instituto Nacional de Estadística y Geografía (INEGI).

Baja California has a water availability of 3250 Mm$^3$ year$^{-1}$. The Valle de Mexicali has water availability of 2950 Mm$^3$ year$^{-1}$, of which 1850 Mm$^3$ year$^{-1}$ is surface water, and

1100 Mm$^3$ year$^{-1}$ is groundwater; the remaining 300 Mm$^3$ year$^{-1}$ is groundwater from the rest of the state and water from dams. The average annual temperature is 22.3 °C, with a minimum of below 0 °C. and a maximum of 50 °C. The warm part of the year is from July to September, and December and January are the coldest months. The average precipitation is 82.9 mm/year and average potential evaporation is 2316 mm year$^{-1}$. The most important economic activity is agriculture, which is characterized by cyclical and perennial crops that make this the most important irrigation region in the country. The major crops are cotton and wheat, followed by corn, sorghum, alfalfa, and asparagus [55].

The Valle de Mexicali has an approximate area of 3709 km$^2$ and is located in a basin of tectonic origin in the seismic zone of the San Andreas Fault. The peninsula is moving away from the continent at a rate of a few centimeters per year [61]. The Valle de Mexicali is bordered on the east by the Colorado River, on the west by the Sierra Cucapá, on the north by the sandy mesa at the border with the U.S., and on the south by the Gulf of California. Because of the lack of precipitation, which averages 100–400 mm annually, 90% of the cultivated area requires mechanized agriculture. An important characteristic of the Valle de Mexicali is its dependence on the Colorado River, whose flow has been diminishing because of climate change and hydrological projects in the U.S. [62].

The Valle de Mexicali aquifer covers an area of 4908 km$^2$; the annual replenishment is 520.5 Hm$^3$, and the volume of the concession is 783.1 Hm$^3$. The aquifer is overexploited by 265.1 Hm$^3$ every year. According to REPDA, 69% of the water in the Valle de Mexicali is surface water and the remaining 31% is groundwater; 85% is for agricultural use, 10% for industrial use, and the remaining 5% for urban public and other uses. As of 31 May 2019, there were 8716 concessions in Baja California, with a total extraction volume of 3717 Hm$^3$ year$^{-1}$, 75% of which is for agriculture, 14% for urban public use, 7% for industry, 3% for energy generation, and the remaining 1% for other uses. The principal source is surface water, which accounts for 62.7% of the total volume extracted, and the remaining 37.3% is groundwater. As of 31 May 2019, the municipality of Mexicali had 1955 concessions, with a total extracted volume of 2513 Hm$^3$ year$^{-1}$, representing 67% of the state's total, and 88.4% of it was used for agriculture, 7.5% for thermoelectric generation, 2.8% for industry, and 1.3% for urban public use [63].

Baja California is one of the states in Mexico facing the greatest challenges in the supply and demand for water, and in public health and the economy [64]. According to CEABC, 2016 [65], there is a permanent process of the overexploitation of water resources, and the deterioration and exhaustion of basins and aquifers is a major problem. The water infrastructure has promoted the management and control of irrigation water and increased the concentration of salts in the low-lying areas of the Imperial Valley and the Valle de Mexicali. Agriculture and the production of geothermal energy have led to changes in the groundwater of the Valle of Mexicali [66]. The contrast in precipitation, ranging from 200–300 mm/year in some areas to 50 mm/year in others [67], accentuates the water problem in Baja California. Water concessions exist in the context of hydrological stress, and the limitations of the supply has provoked growing competition among different users, creating conflicts over its use [65].

## 4. Conflict and Social Participation

### 4.1. Valles Centrales, Oaxaca

The problem in Valles Centrales dates back to the middle of the twentieth century. In September 1967, the administration of President Gustavo Díaz Ordaz (1964–1970) established an indefinite moratorium in the region [64]. Recognizing the chaotic extraction of groundwater and its overexploitation, it permitted extraction for domestic use, but required a permit for all other uses based on a feasibility study (Art. 3, [68]). The president's order also considered the possibility of damage to groundwater reserves, and ruled that all such uses would be regulated (Art. 4d, [68]). In 2005, in this regulatory context, the problem in Valles Centrales was exacerbated by an abnormally dry period [69]. This event contributed

to Valles Centrales becoming highly dependent on groundwater resources; compounded by previous government decisions, the situation worsened.

The conflict erupted that year when the National Water Commission (CONAGUA) decided to activate the 1967 moratorium order after detecting high levels of water extraction. It levied fines against small farmers for excessive extraction after the Federal Electric Commission (Comisión Federal de Electricidad, CFE) reported a marked increase in electricity use in the region. CONAGUA concluded that the electricity use was related to an increase in the volume of water extracted, but it was actually the result of people digging deeper wells in response to a significantly reduced level of precipitation. The people there were unaware of the existence of the moratorium or the restrictions on levels of extraction [70,71] as the government had never discussed them with the indigenous communities in the regulated areas. When these communities learned of the moratorium and the sanctions imposed by CONAGUA and CFE, and of the effects on their livelihood, they organized to protest the implementation of a longstanding policy that was incapable of responding to the situation of the natural resources with respect for indigenous communities. The Coordinating Committee of Peoples United for the Protection and Defense of Water (Coordinadora de Pueblos Unidos por el Cuidado y la Defensa del Agua, COPUDA) was formed to represent 16 indigenous communities and their members. In 2008, COPUDA initiated a legal process involving several lawsuits, with the objective of receiving an exemption from the 1967 moratorium and beginning a process of consultation. The communities sought a change in the moratorium, hydrological studies of the aquifer, and a recognition of the territorial rights of the Zapotec communities of the microregion, with the ultimate goal of assuring the present and future needs of the community.

During the administration of President Enrique Peña Nieto (2012–2018), CONAGUA began a technical study updating the hydrological, social, economic, and environmental conditions of the Valles Centrales aquifer. In 2013, an administrative court, the Primera Sala Regional Metropolitana del Tribunal Federal de Justicia Fiscal y Administrativa, ruled in favor of COPUDA and recommended a change in the 1967 moratorium. Its major finding expressed the need to reconcile the law with the territorial rights of the indigenous communities of the region; not to do so would be a violation of their human rights as established under the Constitution [Arts. 2 and 27, [72]) and Agreement 169 (Arts. 1, 13, and 15) of the International Labor Organization (ILO).

After a lengthy period of consultation and deliberation, which included 32 informational meetings and 11 working meetings in 2015–2019, one result of which was a call for the elimination of the moratorium, was an order that established the Valles Centrales aquifer as a regulated zone [72]. The order, published on 23 January 2020, recognizes the right of indigenous communities to participate in the administration and preservation of national water resources, in accordance with the Constitution, the ILO agreement, and the National Water Law. The order, signed by President Andrés Manuel López Obrador, who took office in 2018, recognizes that Mexico is a pluricultural nation originally made up of its indigenous peoples, and that these communities have the right to free determination in the conservation and improvement of their environment, the preservation of their lands, and the use and enjoyment of the natural resources in the places where they live. It also recognizes the right to the access, availability, and purification of water for personal and domestic use, in a manner that is sufficient, healthy, acceptable, and affordable. It reaffirms the country's commitment to respecting ILO Agreement 169 regarding tribal and indigenous peoples in independent countries. It recognizes indigenous communities as possessors of human rights, implying an obligation by the Mexican state to fully respect and apply them in accordance with Article 1 of the Constitution. It includes a specific provision for the participation of indigenous communities in the administration and preservation of national groundwater, and it recognizes their right to be consulted and to participate in the administration of water and the protection of the aquifer, establishing joint obligations with CONAGUA [73,74].

### 4.2. Valle de Mexicali, Baja California

The arrival in Mexicali of Constellation Brands was part of the company's plans for expansion and took place in the context of a public policy to attract investment to Baja California. The company's business plan for 2012–2018 included an investment of USD 4.6 billion in production plants on Mexico's northern border to satisfy the demands of the U.S. beer market: the Nava plant, in Coahuila de Zaragoza, the Obregón plant, in Ciudad Obregón, Sonora, and the Mexicali plant [75]. The state of Baja California promoted a public policy in 2015–2019 to promote direct foreign investment that included legislation, logistical and water infrastructure, strategic water projects, and media promotion in an attempt to attract a USD 1.4 billion investment in a brewery project in Mexicali [76]. However, government support for this project was provided in the context of public concern that had erupted in 2017 in a social and environmental movement against the privatization of water.

Baja California had thus witnessed various mobilizations since 2017 that presented a variety of demands related to water management [77], including the water concessions to Constellation Brands in Mexicali [31,78,79]. The brewery project planned to occupy 388.5 hectares previously used for agriculture and livestock. There were approximately 800 wells on this land, of which 640 were part of Colorado River Irrigation District 014 and 160 of which were private wells. Their depth ranged from 65 to 180 m, and most yielded from 100 to 160 L/s. However, the Valle de Mexicali aquifer, where the land was located, was under a type 3 moratorium on groundwater extraction [80].

In broad terms, this conflict between business organizations, which supported the brewery, and social organizations, which opposed it, was an obvious one. The construction of the brewery had the support of the federal and state governments until changes in administration in 2018 and 2019, respectively. The new administrations supported social organizations, just as they had promised in their electoral campaigns. The Government Secretary organized a referendum, and the people voted against the brewery project in Mexicali.

The company's arguments revolved around the idea of compliance with the laws relating to its project. It would invest more than USD 1.4 billion, and at maximum production would use less than 0.3% of the total annual volume of water in the Valle de Mexicali. The company argued that its project had complied with the process of obtaining licenses and permits from the federal, state, and municipal authorities, including approval from the federal courts. The company announced that the brewery would use 1.75 $Mm^3$ $year^{-1}$ of water to produce 5 $Mhl$ $year^{-1}$ of beer. The arguments of citizens' associations, under the umbrella organization *Mexicali Resiste*, can be summarized as a concern that Constellation Brands would put the availability of water in the city of Mexicali at risk. Jesús Filiberto Rubio, representing the citizens' groups, filed an application with the Baja California State Electoral Institute (Instituto Estatal Electoral del Estado de Baja California, IEEBC) on 11 October 2018, calling for a referendum on the governors' approval of the company's environmental impact statement.

In brief, the citizens' organizations argued that the project is highly consequential, since it would be located in a desert region without rainfall. Overexploitation of the aquifer could generate saline intrusion; the use of the water would be highly wasteful and could compromise the welfare of future generations. A critical point in their argument is that water is the main ingredient in the production of beer, not a complementary ingredient. They also emphasize that at times when the brewery would be at full capacity, it would be using more water than all other industries in Mexicali and Tijuana combined. They also point out that the only thing about the beer that would be Mexican would be the water and the use of local workers. The question they wanted put to a vote was: "Do you agree that this government should authorize the Constellation Brands Corporation to construct and operate a brewery in Mexicali using water from Baja California?".

The IEEBC [81] received 34 citizen statements related to the referendum between 17 October 2018, and 18 January 2019, of which 14 supported the referendum and 20 opposed. Among those in favor were the *Sección Guerrero de Agua para Todos* (Fighting Division of Wa-

ter for All), the *Frente Social para la Soberanía Popular* (Social Front for Popular Sovereignty), and various other citizens' groups. Those against were organizations related to economic development in the region, including the *Consejo de Desarrollo Económico de Mexicali* (Economic Development Council of Mexicali), Consejo Coordinador Empresarial (Business Coordinating Council), and trade union groups such as the *Liga de Choferes de Mexicali* (League of Mexicali Drivers). Those in favor of the referendum emphasized that "carrying out the referendum is of critical importance," while those opposed demanded that "this inadmissible application for a referendum be analyzed and resolved".

On 4 March 2019, the IEEBC ruled the application inadmissible [82], and on May 29 the Federal Electoral Court rejected the appeal of the citizens' groups [83]. Following this ruling, the citizens' organizations went to the media to appeal for intervention by the federal government, recalling a campaign promise of then-presidential candidate López Obrador in 2018. On 23 September, López Obrador, now president, called for the company to locate its brewery in southern Mexico, where there is a greater availability of water.

With the support of the federal government through the Government Secretary, a referendum took place on 21–22 March 2020, and 36,781 votes were cast, of which 76.1% opposed the construction of the brewery. The options on the ballot were: "I agree that construction should be completed on the Constellation Brands brewery in Mexicali, because the company has already made an investment and created jobs without affecting the community's water supply" or "I disagree that construction should be completed on the Constellation Brands brewery in Mexicali, because I do not want water used for this type of industry." Based on the outcome, the Government Secretary announced that CONAGUA would deny the brewery's pending permits [84]. Federal, state, and municipal authorities met with the company's representatives and agreed on a period of two years, beginning 18 March 2021, in which the company could carry out an orderly dismantling of the unfinished plant [85].

## 5. Discussion and Conclusions

The current study presents lessons and suggests future lines of research for the cases of Valles Centrales and the Valle de Mexicali. From a broad perspective, increasing competition for water resources is a cause of conflicts at different intensities and scales. Communities, municipalities, states, and even transboundary regions are witnessing new challenges and threats to water security, human wellbeing and environmental protection [86–94]. Within this framework, it is important to point out three arguments. First, the success of citizen participation originating from class alliances, where different protesters, as historically excluded actors, decide to mobilize together in order to avoid the implementation of a groundwater policy [95]. Second, formal and transparent processes of citizen participation are key to effective governance and water conflict prevention Cortez [78]. Third, collaboration is the best way to overcome water conflicts, as highlighted by Boelens [96].

The regulation of water resources in Mexico favors the involvement of citizen participation in the decision-making process [97,98]. The social relations established around the issue of groundwater, including its access, use, and exploitation, are heterogeneous across different scales of time and space [99]. The socioenvironmental and cultural conflicts are best understood by those who live in direct contact with the resource in the territory where it is found [100,101]. Conflicts over water are characterized by a common triggering factor of the centralized exercise of power, against which background citizens perceive socioenvironmental damage to their surroundings [102]. Citizen initiatives emerge through social participation, by proposing alternatives and solutions to common problems. Citizen participation represents the only possible path for advancing and innovating the management of water; for mediating the relations of power among communities, the state, and water resources; and, for implementing the Constitutional rights to water and sanitation and to a healthy environment throughout the country. The current paradigm of water management in Mexico must overcome the conflicts related to scarcity and inequality

and must favor the participation of users organized in the search for a consensus on the recognition of their rights to water [26,103–107].

From the specific Mexican experiences cited here, the construction of a hydrosocial territory in the Valles Centrales and Valle de Mexicali aquifers is concretized through the potentiality and significance of citizen participation which favors socially responsible and sustainable groundwater policy. The case of Valles Centrales reflects a major opportunity by which to innovate the management of groundwater resources. The current approach to water resource management should be adapted according to the availability of water from surface and subterranean sources, population density, and the uses of the resource [26]. Government discourse based on sustainable development has no validity if a constant restriction on access and the use of a vital resources places the survival of human populations at risk. For this discourse to be acceptable, it is necessary to adapt the terms of access, use, and exploitation of resources in due time and proper course. The case of Valle de Mexicali shows that political power is necessary for citizens to be able to exercise their rights to organization and participation. The position of the government is essential to an organizations' ability to defend the human right to clean water and sanitation through the prioritization of the use for the urban public over that of industry. Social participation was an effective source of opposition to the brewery, affirming that the water of the Valle de Mexicali aquifer is increasingly scarce and that urban public use should be a priority.

On the map of the political geography of conflicts over Mexico's groundwater, both of these case studies stand out [31]. Valles Centrales and Valle de Mexicali have been described in recent studies as areas of groundwater concentration and of an unequal struggle for water access [27,59,78]. Ibarra-García and Talledos-Sánchez [108] warn that cases such as Mexicali reflect how the formal and informal concentration of groundwater has contributed to a predominance of the economic logic of capital, and of the production, circulation, and consumption of commodities whose processes use water, over other uses. In this context, the indigenous communities of Valles Centrales perceive how institutionality seeks to impose regulatory frameworks that conflict with their traditional environmental practices [27], while the people of Valle de Mexicali oppose the economic logic of business and public policy of promoting investment [76,78]. However, in both cases, citizen participation emerges as a response to government decisions that conflict with the protection and conservation of groundwater.

The contribution of our study to the practice of public policy is twofold: (1) its visibilization of the sociopolitical challenge of the use of, access to, and exploitation of groundwater; and (2) its recognition of citizen participation in the redesign and implementation of public policy for water management.

Finally, to conclude, it is worth to suggest that some of the management recommendations for other regions of the world could rely on the implementation of a robust water policy system, focusing on water use, water conservation and citizen participation as an integrated means by which to sustain water resources at the local, national, regional and international scales. In this case, it would therefore be mandatory to build the foundations for a sense of common responsibility and solidarity within the waterbody and between all different stakeholders, with respect for the environment and human rights.

**Author Contributions:** Conceptualization, O.R.M.-P. and L.F.N.; Formal analysis, O.R.M.-P., L.F.N. and A.C.-C.; Investigation, O.R.M.-P., L.F.N. and A.C.-C.; Methodology, L.F.N.; Writing—original draft, O.R.M.-P., L.F.N. and A.C.-C.; Writing—review and editing, L.F.N. All authors have read and agreed to the published version of the manuscript.

**Funding:** This research received no external funding.

**Data Availability Statement:** Not applicable.

**Acknowledgments:** The authors would like to thank the editors and anonymous reviewers for their comments and suggestions, which helped in making this paper better.

**Conflicts of Interest:** The authors declare no conflict of interest.

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
