# Peer review of "The Visibility of Citizen Participation and the Invisibility of Groundwater in Mexico"

_water, doi:10.3390/w14091321_

Round 1

Reviewer 1 Report

The paper requires significant improvements, especially in the description of the research methods.  Please include a research flow chart of the procedures performed.  Please describe precisely how the source materials were selected? According to what criteria was an extended content analysis performed? What were the criteria for textual material queries?  Please describe very precisely the social research methods used in this study. Please discuss the relevant literature, in particular relate your methods to those used by other researchers.  Please develop a discussion of the theoretical basis adopted in the analyses. The principle of "subjective interpretation" adopted by the authors, in my opinion, does not meet the rigours of the scientific method. Please propose a logical framework for the interpretations carried out. The source materials should be updated with the latest available work and discussed accordingly. 

Author Response

Editor’s Overall Response

Dear Dr. Nava,

Manuscript ID: water-1653941

Title: The Visibility of Citizen Participation and the Invisibility of Groundwater in Mexico

Authors: Ojilve Ramón Medrano-Pérez, Luzma Fabiola Nava *, Antonio Cáñez-Cota

Received: 10 March 2022

E-mails: omedrano@conacyt.mx , nava@iiasa.ac.at , acanez@conacyt.mx

Submitted to section: Water Use and Scarcity,

https://www.mdpi.com/journal/water/sections/Water_Use_Scarcity

Water Conflict Prevention

https://www.mdpi.com/journal/water/special_issues/Water_Conflict

Manuscript Status: Pending major revisions

You can find your manuscript and review reports at this link:

https://susy.mdpi.com/user/manuscripts/resubmit/a6b3b3896080d42a3cf07e1152bdb2cf

Thank you in advance for your kind cooperation and we look forward to hearing from you soon.

Kind regards,

Dolphin Dou

Assistant Editor

E-Mail: dolphin.dou@mdpi.com  

General response to the Editor

Dolphin Dou

Assistant Editor

Dear Dolphin Dou,

The authors have revised the manuscript with an overall objective to keep the description focused on relevant and essential information needed to discuss the topic of citizen participation in Mexico groundwater policy making.

The authors have addressed the reviewer comments, keeping and improving the original idea of the manuscript. I should emphasize that the authors did their best to provide essential content while addressing the reviewer’s suggestions. By doing that, we hope that all the edits address not only the comments of the reviewers but also keep the content focused.

Thank you in advance.

Response to Anonymous Reviewer 1

Dear Reviewer 1,

Thank you for your comments. The authors have strived for including all the relevant information related to citizen participation over groundwater resources, while at the same time being brief and precise in the content. The authors have revised the manuscript with an overall objective to keep the description focused on relevant and essential information needed to discuss the topic of citizen participation in Mexico. The authors have revised the manuscript and addressed the reviewer comments, keeping the aim of the manuscript intact. We hope we have responded constructively to your comments and suggestions.

The following are our answers to your comments. Please note that underlined text responds to our revisions in the document.

***

Reviewer: “The paper requires significant improvements, especially in the description of the research methods.”

Authors’ Response: Thank you for taking the time to review our manuscript, for the positive feedback and comments, all of them have greatly improved the content of the paper. We have improved the paper according to yours and all reviewer’s comments, we hope we addressed your comments. We hope that in this revised version the paper is better suitable to all audiences, so everyone can benefit from its content and recommendations. In this order of ideas, to better show the edits to the manuscript, we have add a comment indicating to which of your comments/suggestions we are our addressing. The response to this general comment can be found at the section : 3. Materials and Methods. The whole section 3 has been revised, and while doing that, we have also responded to the particular comments: “Please describe very precisely the social research methods used in this study” and “Please describe very precisely the social research methods used in this study.”

Reviewer: “According to what criteria was an extended content analysis performed?” and “What were the criteria for textual material queries?”

Authors’ Response: Thank you for your comments. The authors addressed these comments in a very comprehensive way. We describe the criteria of the content analysis method in relation to citizen participation:

This study used qualitative methods to recognize the significance of citizen par-ticipation in the process of decision-making and innovations in water policy. Social re-lations matter, and the words and actions of the actors studied are of great importance. A review of the literature relative to social participation and conflict surrounding groundwater management and in Mexico was the focus of a descriptive and contextual approach based on what stakeholders to collectively say to be the source of the con-flictual situation, and what they wish for the future of their relation to water resources in the short, medium and long terms.

Reviewer: “Please describe precisely how the source materials were selected?”

Authors’ Response: further details are giving and read as follows:

Detailed information (scientific, technical, and official documents, as well as online newspaper archives), based on the most recent information, accountability, and coherence, was collected and analyzed to develop a holistic vision of the object of study [36].

Reviewer: “The principle of "subjective interpretation" adopted by the authors, in my opinion, does not meet the rigours of the scientific method.”

Authors’ Response: To solve this comment we have included, and explained, a very important characteristic of the qualitative method.

Nevertheless, the methodological limitations of the study translate into the illus-tration of events through sources of information . This means that, even if qualitative data cannot be objectively measured or counted, it expresses the interpre-tive qualities of an event. Then this event, associated with ideas, opinions, values, and behaviours of individuals, allows to the researcher to explore and provide deeper in-sights into real-world problems. This said, a qualitative method has been used to un-derstand people's beliefs, experiences, attitudes, behavior, and interactions with the aim of looking at meaning, perspectives and motivations [47].

In fact, after having adressing your comments, we have updated the Abstract; it now reads as follows:

Abstract: The aim of this study is to assess the social responses to protect and prevent conflict over groundwater resources. By means of a qualitative method and the study of the Valles Centrales and Valle de Mexicali aquifers in Mexico, we have found that centralized water management, citizens’ socio-environmental awareness, asymmetry of power between stakeholders, imposition of government policies, and economic interests are all factors present on emerging conflicts over groundwater. However, citizen participation develops to provide to, organized individuals, an opportunity to influence public decisions through the recognition of their rights with respect to water inequalities. However, a limitation of the study consists on the illustration of conflictual events through the interpretation of qualitative data and of the opinions of the actors studied. However, the construction of hydrosocial territory in these aquifers is concretized in the potentiality and significance of citizen participation in promoting sustainable and socially responsible public groundwater policy at the regional level.

Reviewer: “Please develop a discussion of the theoretical basis adopted in the analyses.”

Authors’ Response: Thank you for this comment. The whole section 5 has been revised in order to solve this importat comment. In concrete terms, the very first paragraph has been edited and robusted with some scientific related references. Now it reads as follows:

The current study presents lessons and suggests future lines of research sur-rounding the cases of Valles Centrales and the Valle de Mexicali. From a broad per-spective, increasing competition for water resources is causing conflicts at different intensities and scales. Communities, municipalities, states, and even transboundary regions are witnessing new challenges and threats to water security, human wellbeing and environmental protection [86 – 94]. In this order of ideas, it is important to point out three arguments. First, the success of citizen participation coming from class alliances, where different protesters, as historically excluded actors, decide to mobilize together in order to avoid the implementation of a groundwater policy [95]. Second, formal and transparent processes of citizen participation are key to effective governance and water conflict prevention [Cortez [78]. Third, collaboration is the best way out of water con-flicts Boelens [96].

Reviewer: “The source materials should be updated with the latest available work and discussed accordingly.”

Authors’ Response: Thank you for your comment. The authors have improved the paper according to the reviewer’s suggestions. We hope the reviewer’s comments have been addressed. Regarding this specific comment, we should mention that the whole document has been revised; therefore, the References section has been updated with the new references integrated in this revised version. Please note the new references [44- 47; 86 – 95] added to support the revisions we have accordingly made.

Reviewer 2 Report

Groundwater management is essential in many regions, especially in countries with a dry climate. The risk of centralized management without taking into account the local population is a major problem when it comes to resources as fundamental as water. This paper deals with two cases of local involvement in groundwater management, specifically in Mexico, making extensive use of bibliography. In general, it is a very detailed work, but it needs to improve some details:

  • The abstract is very short and could be expanded by providing more information on the cases analyzed and the results of the paper.
  • In the discussion and conclusions section, it would be good to emphasize the management recommendations for other regions of the world, based on the two case studies analyzed. In addition, it will be good to provide proposals for future studies along the lines of this work, and expose the limitations of the work (something mentioned in the abstract but not in the article).

Other minor items are:

  • The figures are a bit small and difficult to see. They could be the width of the sheet.
  • The format of citations and references should be reviewed.

Author Response

Editor’s Overall Response

Dear Dr. Nava,

Manuscript ID: water-1653941

Title: The Visibility of Citizen Participation and the Invisibility of Groundwater in Mexico

Authors: Ojilve Ramón Medrano-Pérez, Luzma Fabiola Nava *, Antonio Cáñez-Cota

Received: 10 March 2022

E-mails: omedrano@conacyt.mx , nava@iiasa.ac.at , acanez@conacyt.mx

Submitted to section: Water Use and Scarcity,

https://www.mdpi.com/journal/water/sections/Water_Use_Scarcity

Water Conflict Prevention

https://www.mdpi.com/journal/water/special_issues/Water_Conflict

Manuscript Status: Pending major revisions

You can find your manuscript and review reports at this link:

https://susy.mdpi.com/user/manuscripts/resubmit/a6b3b3896080d42a3cf07e1152bdb2cf

Thank you in advance for your kind cooperation and we look forward to hearing from you soon.

Kind regards,

Dolphin Dou

Assistant Editor

E-Mail: dolphin.dou@mdpi.com  

General response to the Editor

Dolphin Dou

Assistant Editor

Dear Dolphin Dou,

The authors have revised the manuscript with an overall objective to keep the description focused on relevant and essential information needed to discuss the topic of citizen participation in Mexico groundwater policy making.

The authors have addressed the reviewer comments, keeping and improving the original idea of the manuscript. I should emphasize that the authors did their best to provide essential content while addressing the reviewer’s suggestions. By doing that, we hope that all the edits address not only the comments of the reviewers but also keep the content focused.

Thank you in advance.

Response to Anonymous Reviewer 2

Dear Reviewer 2,

Thank you for your comments. The authors have strived for including all the relevant information related to citizen participation over groundwater resources, while at the same time being brief and precise in the content. The authors have revised the manuscript with an overall objective to keep the description focused on relevant and essential information needed to discuss the topic of citizen participation in Mexico. The authors have revised the manuscript and addressed the reviewer comments, keeping the aim of the manuscript intact. We hope we have responded constructively to your comments and suggestions.

The following are our answers to your comments. Please note that underlined text responds to our revisions in the document.

***

Reviewer: “The abstract is very short and could be expanded by providing more information on the cases analyzed and the results of the paper.”

Authors’ Response: Thank you for the comment. The authors are glad to hear that our research work is recognized and well valued. The authors have revised the entire manuscript with an overall objective to keep the description focused on relevant and essential information needed to discuss the topic of citizen participation related to groundwater resources. The revised version of the Abstract reads as follows:

Abstract: The aim of this study is to assess the social responses to protect and prevent conflict over groundwater resources. By means of a qualitative method and the study of the Valles Centrales and Valle de Mexicali aquifers in Mexico, we have found that centralized water management, citizens’ socio-environmental awareness, asymmetry of power between stakeholders, imposition of government policies, and economic interests are all factors present on emerging conflicts over groundwater. However, citizen participation develops to provide to, organized individuals, an opportunity to influence public decisions through the recognition of their rights with respect to water inequalities. However, a limitation of the study consists on the illustration of conflictual events through the interpretation of qualitative data and of the opinions of the actors studied. However, the construction of hydrosocial territory in these aquifers is concretized in the potentiality and significance of citizen participation in promoting sustainable and socially responsible public groundwater policy at the regional level.

Reviewer: “The figures are a bit small and difficult to see. They could be the width of the sheet.”

Authors’ Response: Thank you for your comment, the three figures included in the manuscript have been widened to improve their visibility.

Reviewer: “[…] To expose the limitations of the work (something mentioned in the abstract but not in the article)” and “In addition, it will be good to provide proposals for future studies along the lines of this work”

Authors’ Response: Please note that the whole section 3 has been edited accordingly to all reviewers’ comments. However, these two comments are addressed at the end to section 3, and both read as follows:

The methodological contribution of the study is twofold. First, it examines the patterns in the behavior of citizen participation regarding conflicts over groundwater. Second, it contributes to the debate about water policy with evidence obtained from the case studies, by objectively identifying the facts just as they presented themselves. In this order of ideas, it is important to highlight that two main roots of methodologi-cal strategies in the literature of citizen participation have been identified. First, the evaluation of effectiveness of citizen participation; that is, research focus on explaining to what extent stakeholders improve societal outcomes and achieve public value [44]. Second, the searching of the intersection between citizen participation and policy in-novation, focusing in agenda settings [45]. That is, increasing legitimacy in deci-sion-making processes towards collective learning, since some democracies have weak institutions to enforce accountability [46]. We stand in this last root, since our method-ological strategy is based on describing citizen participation to block imposed water policies and promote innovation through opening decision making to general participation.

Nevertheless, the methodological limitations of the study translate into the illus-tration of events through sources of information . This means that, even if qualitative data cannot be objectively measured or counted, it expresses the interpre-tive qualities of an event. Then this event, associated with ideas, opinions, values, and behaviours of individuals, allows to the researcher to explore and provide deeper in-sights into real-world problems. This said, a qualitative method has been used to un-derstand people's beliefs, experiences, attitudes, behavior, and interactions with the aim of looking at meaning, perspectives and motivations [47].

Therefore, we propose a future re-search agenda conducting in-deep interviews and focus group to use them in a more comprehensive qualitative research design and learn the sense of the action of the ac-tors involved. This strategy will help to identify subjectivities that allow discussion of the results in a more complex and robust way; and, to learn more about causal processes in citizen participation in groundwater projects

Reviewer: “it would be good to emphasize the management recommendations for other regions of the world, based on the two case studies analyzed. … In addition, it will be good to provide proposals for future studies along the lines of this work”

Authors’ Response: Thank you for your comment. Please note that the whole section number 5 has been revised according to your comments and suggestions. However, in a specific manner, we have included a new argument; and it reads as follows:

Finally, to conclude, it is worth to suggest that some of the management recom-mendations for other regions of the world could rely on the implementation of a robust water policy system, including water use, water conservation and citizen partici-pation as an integrated means to sustain water resources at the local, national, regional and international scales. In this case, would therefore be mandatory, to build a com-mon responsibility and solidarity within the waterbody and between all different stakeholders, including respect for the environment and human rights. 

Reviewer: “The format of citations and references should be reviewed.”

Authors’ Response: Thank you for your comment. The authors have improved the paper according to the reviewer’s suggestions. We hope the reviewer’s comments have been addressed. Regarding this specific comment, we should mention that the whole document has been revised; therefore, the References section has been updated with the new references integrated in this revised version. Please note the new references [44- 47; 86 – 95] added to support the revisions we have accordingly made.

Round 2

Reviewer 1 Report

Paper is improved. Thank You.